# Enhancing Ensemble Learning Using Explainable CNN for Spoof Fingerprints

**DOI:** 10.3390/s24010187

**Published:** 2023-12-28

**Authors:** Naim Reza, Ho Yub Jung

**Affiliations:** Department of Computer Engineering, Chosun University, Gwangju 61452, Republic of Korea; naim.dev.service@gmail.com

**Keywords:** ensemble learning, convolutional neural network, class activation map, fingerprint, spoof detection

## Abstract

Convolutional Neural Networks (CNNs) have demonstrated remarkable success with great accuracy in classification problems. However, the lack of interpretability of the predictions made by neural networks has raised concerns about the reliability and robustness of CNN-based systems that use a limited amount of training data. In such cases, the utilization of ensemble learning using multiple CNNs has demonstrated the capability to improve the robustness of a network, but the robustness can often have a trade-off with accuracy. In this paper, we propose a novel training method that utilizes a Class Activation Map (CAM) to identify the fingerprint regions that influenced previously trained networks to attain their predictions. The identified regions are concealed during the training of networks with the same architectures, thus enabling the new networks to achieve the same objective from different regions. The resultant networks are then ensembled to ensure that the majority of the fingerprint features are taken into account during classification, resulting in significant enhancement of classification accuracy and robustness across multiple sensors in a consistent and reliable manner. The proposed method is evaluated on LivDet datasets and is able to achieve state-of-the-art accuracy.

## 1. Introduction

Ensembling of neural networks to enhance their accuracy and robustness has been a well-established concept since 1990 when it was first introduced by [1]. Subsequently, numerous ensemble techniques have been developed, such as the cross-validation ensemble [2], model averaging ensemble, weighted average ensemble, horizontal and vertical ensemble [3], which are notable among others. While these methods have achieved significant success at enhancing the accuracy of a model’s predictions, they are also associated with a large training cost. To address this issue, the authors of [4] proposed a method that uses a cyclic cosine annealing learning rate, as proposed by [5], to guide a neural network towards different local minima and to save the weights of the network at the end of each cycle. This approach produces diverse ensemble members under a single training session. Furthermore, in [6], the authors introduced Stochastic Weight Averaging, a method that forms an ensemble in the weight space by implementing a moving average of the weights of the models, as opposed to averaging the outputs of the models. These ensemble learning methods provide a convenient approach to enhance a network’s performance and robustness.

In security applications such as detecting spoof biometrics, maintaining the robustness of the network is crucial. While the ensemble method offers potential to enhance network robustness, it often involves various trade-offs. As depicted in Figure 1, implementing the snapshot ensemble method [4] in conjunction with the cyclic cosine annealing learning rate [5] for all four sets of sensor data present in the LivDet [7] dataset leads to the degradation of accuracy in a network that had already been producing state-of-the-art results. This observation suggests that in cases for which the training dataset is small and homogeneous in nature, the number of available local minima is very limited, and it is challenging to ensure diversity among the ensemble members. The availability of biometric datasets, such as a fingerprint dataset for training a spoof detection network, is limited due to the sensitive nature of the data. The small size of the dataset can introduce significant biases during training, which may cause a reduction in ensemble accuracy.

Explanation of the learned features of a network is instrumental to investigate potential failure modes resulting from biases in the dataset. Additionally, the ability to explain the functioning of a spoof detection network is imperative to enhance the system’s reliability. Recognizing the complexity associated with interpreting CNNs, various methodologies have been proposed, such as the Class Activation Map (CAM) [8] and the Gradient-weighted Class Activation Map (Grad-CAM) [9], to identify and visualize the specific image regions that the network utilizes for prediction. Thus, the explanations are mainly limited to the production of a saliency map, and the internal representation of the CNN is still mostly unexplainable. Furthermore, the produced saliency maps are specific for each sample, and the global interpretation of the network or the dataset is not easily found.

Nevertheless, sample-wise saliency maps can be effective information for training new networks that can use different regions for classification. In our approach, we have leveraged the power of CAM to extract explanations from a trained network regarding its prediction and utilize the saliency map to steer the learning of another network. By leveraging these saliency maps, we can refine our networks to focus on other relevant features, potentially reducing the influence of biases and improving the overall efficacy of the spoof detection system.

After analyzing the class activation map of a previously trained model, we have observed that certain areas of the input image are entirely disregarded by the network for classification, despite containing critical features, as demonstrated in Figure 2. Additionally, strong activation regions are relatively small, which indicates that the network may ignore important parts of the input image, which could include essential information such as textures or patterns. This observation also suggests that activation perturbation of a network can be utilized to generate diverse models that can serve as ensemble members to enhance the network’s robustness and accuracy.

Though spoof fingerprints can be easily created using low-cost and readily available materials such as wood glue, Play-Doh, gelatin, and latex-like substances [7,10] to deceive a biometric authentication system, fabricating such spoof fingerprint can be time consuming. As a result, the datasets available for training a spoof detection network are comparatively very small in size.

In our study, we begin by training a CNN to generate a response map of a fingerprint using the network architecture described in Table 1. The input fingerprint size is 512×512, and the size of the generated response map is 26×26. We use this network to generate activation maps for given input images. Then, we select the most-active regions of the input image and add random noise to those regions. The resulting image is used as the input to train another network that shares the same architecture as the base network and has a constant learning rate. We produce multiple networks using our training method, and afterward, we ensemble the resulting networks with the base network. Our method ensures different convergence points by using the activation map to hide selected regions of the input, resulting in perturbation of activation in the network and thereby producing diverse models. An instance of the change in activation after training with the proposed method is illustrated in Figure 2c. Our method significantly improves the robustness and accuracy of the network compared to traditional ensembles by taking advantage of class activation maps.

In our evaluation and training, we utilized the LivDet 2015 dataset, which comprises testing and training sets for four scanners, as mentioned in [7]. The training sets were employed for network training purposes, while the testing sets were employed for network evaluation. We evaluated our method against previous approaches using all the testing sets.

To summarize, our contributions are as follows. We demonsrate that activation perturbation of a network can be utilized to enhance ensemble networks by ensuring model diversity through the use of class activation maps, as described in Section 3. This approach ensures that the ensemble networks use most of the feature regions in a fingerprint during inference while providing transparency regarding the networks’ decision-making process. The proposed training method enhances the accuracy and robustness of an ensemble network for detecting spoof fingerprints. This approach is particularly designed to mitigate the issue of performance degradation in ensemble learning caused by limited training datasets and fewer good local minima. The proposed method is evaluated against recent methods in Section 5.2, where it outperforms the state-of-the-art methods.

The subsequent sections of this paper are arranged in the following manner. We begin by discussing previous works that have provided inspiration for our research in Section 2. Following this, we introduce our proposed methodology in Section 3. To demonstrate the implementation of our method, we utilize the network architecture presented in Table 1 and elaborate on the implementation process in Section 4. In Section 5, we present a comparative analysis between our proposed method and state-of-the-art techniques. Finally, in Section 6, we discuss the impact of our proposed method and outline future research directions.

## 2. Related Work

Using an ensemble of neural networks offers a simple yet effective measure to improve performance and robustness beyond that of a single network [11,12,13]. Notably, in high-profile competitions such as ImageNet [14] and Kaggle (www.kaggle.com, accessed on 26 November 2023), ensembles of deep learning architectures have consistently outperformed individual models. Prior studies have demonstrated that ensembling can enhance both accuracy and robustness by exploiting network diversity [15,16,17,18,19,20,21]. Despite these benefits, the use of ensembling for neural networks remains limited due to high training costs. In light of this problem, Snapshot Ensemble [4], Fast Geometric Ensemble [22], and Stochastic Weight Averaging ensemble [6] have been proposed, wherein the authors exploit model diversity and geometric properties of the loss surface to achieve the benefits of ensembling. However, these techniques have mainly been evaluated on relatively large datasets, wherein at least 50,000 images are available in the training set, and there has been limited exploration of their effectiveness on smaller datasets such as LivDet 2015 [7].

In this work, we have addressed the issues associated with ensemble learning when using very small datasets for spoof detection in fingerprints. To overcome these challenges, we have leveraged Grad-CAM [9] to create an explainable CNN that generates activation maps [23]. Grad-CAM has been successfully applied to explain classifiers in image classification, image segmentation [24], and visual question answering (VQA) [25]. Its success has led to the development of Grad-CAM++ [26], which further enhances the explanation capabilities of Grad-CAM and has been used for object detection and localization [27,28,29]. The authors in [30] employed Grad-CAM as a visualization tool to identify and highlight noise across various channels of a network when processing a fingerprint image. The use of CAM is also presented in the study by [31], where it was used for patch extraction during the inference stage. In our work, we have utilized Grad-CAM not only for visual explanation of the network but also to mitigate the biases present in the dataset through that explanation. The activation maps generated by the network serve as a crucial component of our training method by enhancing the ensemble by promoting model diversity.

The detection of spoof fingerprints remains a prominent research topic, and CNNs have proven to be a successful approach [32]. To stimulate further research efforts in this field, several spoof detection competitions (LivDet 2009–2021) have been organized [7,10,33,34]. Notably, in the LivDet 2015 competition, the authors of [35] employed a transfer learning approach using deep CNNs, specifically AlexNet [36] and VGGNet [37], which were pre-trained on the ImageNet [14] dataset, and fine-tuned some of the layers for spoof detection of fingerprints. Meanwhile, [38] proposed a CNN architecture with similar performance but reduced test time. Improving the robustness of CNN-based spoof detection systems has been explored by [39], who adopted a hybrid approach that combines hyper-parameter tuning of a CNN and Support Vector Machines (SVMs). Siamese network architecture has been employed by the authors of [40] to improve the robustness of a fingerprint spoof detection system. The authors of [41] have proposed an altered ResNet architecture to achieve smaller parameter size and computational efficiency for a practical spoof detection application. Additionally, [42] have improved the accuracy and robustness of their system using the MobileNet-V1 [43] architecture in conjunction with the fusion of minutiae-based center-aligned local patches. However, their approach involves several complex algorithms, such as minutiae detection, local patch extraction, and patch alignment, in the training and testing procedures. Several techniques [44,45,46,47] have been proposed to generate synthetic fingerprints by leveraging style-transfer techniques. The primary objective of these approaches is to increase the number of data samples in order to address challenges associated with limited dataset sizes. A brief summary of the relevant studies is presented in Table 2.

Our research demonstrates a significantly simplified approach to improve network performance. The work presented here is inspired by the principles introduced by the study [4], wherein the authors established that successful ensemble models can be achieved through ensuring diversity among ensemble members. Additionally, they demonstrated that diverse ensemble members can be obtained with the same training cost required to train a single network. Our contribution builds upon these findings firstly by showing that model diversity can be accomplished through activation perturbation, a technique that involves generating network explanations aided by a class activation map [9,26], and secondly by leveraging this diversity via ensemble techniques.

## 3. Methodology

In this section, we present our proposed training procedure, which employs the network architecture shown in Table 1. We refer to this network as Spoof Detection CNN. Our objective is to generate activation maps from this network. The input to our network is a grayscale image represented as $x\in R^{1\times W\times H}$, where *W* and *H* are the width and height, respectively, of the input image. The network produces a response map as an output $h\left(x\right)\in R^{1\times u\times v}$, with *u* and *v* representing the width and height, respectively, of the response map. The values of the response map of the CNN are expected to be $h_{ij}\in \left[-1,1\right]$ after training.

Let us say the activation map is $L^{c}\in R^{u\times v}$ of width *u* and height *v* for any class *c*. We first compute the gradient of the response $h^{c}$ for predicting the class *c* with respect to the feature map activation $A^{k}$ of a convolutional layer.
(1)$$\nabla_{k}^{c}=\frac{\partial h^{c}}{\partial A^{k}}$$
The backpropagated gradients are globally averaged and pooled across the network’s width and height dimensions, indexed by *i* and *j*, respectively, leading to the derivation of neuron importance weights $\alpha_{k}^{c}$ [9].
(2)$$\alpha_{k}^{c}=\frac{1}{Z}\sum_{i}\sum_{j}{\left(\nabla_{k}^{c}\right)}_{i,j}$$
Here, $\alpha_{k}^{c}$ signifies the relevance of the feature map *k* to the intended target class *c*. Then, the final activation map is obtained by weighted summation of the activations, followed by a ReLU operation [9].
(3)$$L^{c}=ReLU\left(\sum_{k}\alpha_{k}^{c}A^{k}\right)$$
To identify the most-active regions of the activation map, we propose to apply a threshold *t* on the generated map $L^{c}$, and we obtain $L_{t}^{c}$ by
(4)$$L_{t}^{c}=\left\{\begin{array}{cc} L_{k}^{c} & where,\;L_{k}^{c}>t\\ 0 & otherwise \end{array}\right.$$
Following this, we resize the activation map to match the dimensions of the input image and compare the two, as illustrated in Figure 3. We then alter the pixel values of the input image with random noise where the activation map has positive values and obtain:(5)$$\hat{x}=\left\{\begin{array}{cc} r(.) & where\;{\left(L_{t}^{c}\right)}_{ij}>0\\ x_{ij} & otherwise \end{array}\right.$$
where r(.) represents a random noise generator, and $\hat{x}$ denotes the modified input image. The modified input image $\hat{x}$ is then fed into the network that we want to train. Then, the second network generates its response map, and we calculate the loss via Mean Square Error (MSE) with respect to the ground truth. Stochastic Gradient Descent (SGD) with a constant learning rate is employed as our optimization procedure.

Subsequently, we save the model weights at the end of a training session, and we generate *N* models to use in the final ensemble. The use of random initialization at the start of the training session along with our CAM-guided training procedure ensures the production of *N* models with different weights. We select a subset of *n* models (n≤N) and our pre-trained CNN to create our ensemble network. At test time, the ensemble network takes an input $x\in R^{1\times H\times W}$ and generates (n+1) response maps. The response maps are then stacked to produce the final output of the network: $h_{stacked}\in R^{(n+1)\times u\times v}$, as depicted in Figure 4. Stacking the response maps improves the pixel-wise labeling accuracy of the input image, resulting in increased network robustness and accuracy. Finally, if there are *m* elements in $h_{stacked}$, we obtain the final prediction of the network *p* by computing the mean value of $h_{stacked}$.
(6)$$p=\frac{1}{m}\sum_{i}\sum_{j}\sum_{k}{\left(h_{stacked}\right)}_{ijk}$$
Here, p∈[−1,1], and if the value of *p* is negative, then the image is classified as a spoof, and if the value is positive, then the image is classified as real for the fingerprint spoof detection system.

## 4. Training Implementation

### 4.1. Spoof Detection CNN

In this section, we present an implementation of our proposed training procedure, which employs our Spoof Detection network. The details of the network architecture used are provided in Table 1. This network accepts input in the shape of 1×512×512 and generates a response map in the shape of 1×26×26 as output. The proposed Spoof Detection network offers a significant advantage in its training efficiency, primarily due to its ability to achieve effective learning with a larger output map size while requiring fewer fingerprint samples for training. This efficiency is further enhanced by maintaining a moderate depth in the network architecture, consisting of 5 convolution layers. This makes the network sufficiently deep for feature extraction while avoiding the over-fitting issues often encountered by deeper networks.

A notable aspect of this network is the utilization of a relatively large 7×7 filter in the initial layer. This choice is particularly effective for covering larger strides in the input image. The impact on computational expenses due to the use of a larger filter is mitigated by the fact that the input comprises only one channel. Hence, the computational demand is less pronounced compared to scenarios for which larger filters are employed in subsequent layers with multiple channels. To further optimize the network for practical applications, particularly in terms of computational efficiency, the number of filters in each layer is capped at 32. Additionally, we have avoided the use of fully connected layers with a cross-entropy loss function at the end of the network; instead, we produce a response map.

The response map generated by the Spoof Detection network is a 1×26×26 feature response with values ranging from [−1,1], indicating the label (positive/negative) of the features within the corresponding region of the input image. Positive values in the response map signify the probability of live features being associated with the corresponding regions of the input image, while negative values represent the probability of spoof features. An example of a response map produced by the Spoof Detection network is illustrated in the accompanying Figure 5.

### 4.2. Data Processing

The images used in this paper are grayscale images with a white background and dark foreground produced by four different data modules (i.e., scanners) with varying sizes. To pre-process the images, we first invert the color of the image to produce a zero-background image. Afterwards, we crop the foreground and place it in the center of a 512×512 zero-background image. For images larger than 512×512, this operation serves as a cropping mechanism of the foreground from its center with a 512×512 window, and for smaller images, it serves as a translation of the foreground to the center. Our aim is to avoid any loss of textural information of the fingerprint due to down- or upscaling; hence, we do not resize the images. Additionally, we incorporate data augmentation by randomly flipping the images horizontally and vertically and randomly rotating them by $\left[-30^{\circ },30^{\circ }\right]$ in 50% of the training set. Finally, we min–max normalize the pixel values to be in the [0,1] range before feeding them into the network for training. Some examples of the resulting preprocessed images are illustrated in Figure 6.

### 4.3. Ground Truth

The proposed method utilizes tensors of shape 1×26×26 with all values being equal to +1 for live and −1 for spoof fingerprints as the ground truth for the network. We obtain the ground-truth tensors while sampling images form the dataset. Upon sampling images from the dataset, we first gather all live samples corresponding to a specific sensor. For each of these live samples, a 1×26×26 tensor filled with +1 values is generated and paired with the sample. Similarly, for each spoof sample, we generate a corresponding tensor filled with −1 and pair it with the sample. Once the ground-truth tensors are prepared for both live and spoof samples, we apply a series of data processing steps as outlined in Section 4.2. After the processing of samples, the output for each fingerprint sample includes the processed image, its corresponding ground-truth tensor, and the size of the detected fingerprint. The size of the fingerprint is later used to add noise at the exact position of the fingerprint.

The Mean Square Error is employed as the loss function, and Stochastic Gradient Descent is the optimization procedure. The optimization procedure seeks to minimize the difference at each position of the 1×26×26 map. This setup enables the network to learn to label the input image pixel-wise as live or spoof. Notably, the zero-background present in both live and spoof images is automatically trained to be associated with the neutral zero value in the response map.

### 4.4. Training Parameters

To ensure reasonable variance for the network, we employ Xavier initialization [48] to initialize the weights of the network before starting the training. A batch size of two and a learning rate of 0.01 are used for training the base Spoof Detection network. Training is conducted separately for each data module; the network is exposed to approximately 1000 positive-labeled images (i.e., live fingerprint images) and about 1000 to 1500 negative-labeled images (i.e., spoof fingerprint images) from the LivDet 2015 and LivDet 2017 datasets [7,10]. The network is then trained for around 512 epochs on the LivDet 2015 dataset and for 1000 epochs on the LivDet 2017 dataset, which took approximately one day for all the scanners.

### 4.5. Training Ensemble Members

Upon completing the training of the Spoof Detection CNN, referred to as the “Pretrained CNN” in Figure 3, we employ the network to extract an activation map for a given input image. We apply the aforementioned pre-processing and data augmentation steps to prepare an input of size 1×512×512, and we subsequently feed the input into the base network to generate a response map of size 1×26×26. The activation map is generated from the last “ReLU” operation of the Spoof Detection network by performing a backward pass on the generated response map, as defined in Section 3. We then employ a threshold on the activation map to identify the most-active regions.

The analysis presented in Figure 7 provides insightful observations regarding the impact of threshold settings on the processing of fingerprint images in our study. Specifically, it highlights that when lower threshold values are employed, a substantial portion of the input fingerprint image becomes obscured. This obscuration leads to the loss of essential features, which are vital for the accurate identification of live and spoof fingerprints. In light of the ablation study presented in Section 5.6, we set the optimal threshold value as t=0.9 to achieve the results presented in this study for all the sensors.

Following this, we resize the activation map using bilinear interpolation to match the given size of the input fingerprint, and we compare the input image with the activation map. We replace the pixel values with random noise where the activation map has positive values, resulting in a fingerprint image that contains noise in specific areas. This modified image is then used as the input to our second network, which we aim to train for ensembling. An example of the modified input image is shown in Figure 8. This forces the network to allocate its focus to regions outside of the initial active regions of the input image, leading to perturbation of the network’s activation.

We employ the same ground truth and loss function mentioned previously and use a learning rate of 0.01 to train this network. The weights of the network are saved at the end of the training loop. We use random initialization before starting a new training session and follow the aforementioned procedures to generate a new model. We produce several models with the intention of utilizing them as ensemble members in subsequent stages. The proposed training steps are presented in Algorithm 1.

In the context of binary classification for spoof detection of fingerprints, the number of local minima that can be converged to is limited due to the scarcity of a large dataset. In light of this, our proposed training method follows a similar approach to snapshot learning [4], but instead of using a cyclic learning rate, we employ activation maps. Our final network comprises the base Spoof Detection network and *n* ensemble member networks. It takes an image of size 1×512×512 as input and generates n+1 response maps of size 1×26×26, which are stacked together and utilized for final prediction, as illustrated in Figure 4. Final prediction is achieved by calculating the mean value of the stacked response map.
**Algorithm 1:** Proposed Training Method
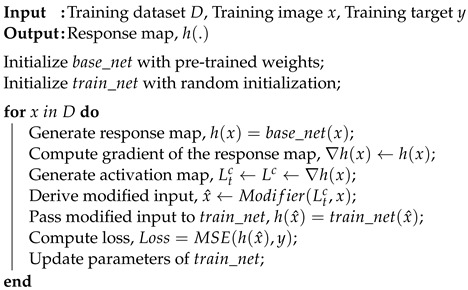


## 5. Experiments

In order to assess the effectiveness of our proposed method, we conducted a series of experiments. The results of these experiments demonstrate a significant improvement in the performance of our ensemble network as compared to previous ensemble methods, as illustrated in Figure 9. Specifically, our proposed approach yielded an overall accuracy increase of 1.51% over the prior state-of-the-art results. Furthermore, our method was able to achieve an overall increase in cross-material robustness by 1.07% compared to previous state-of-the-art results.

### 5.1. Dataset

For our experimental evaluation, we utilized the following datasets.

#### 5.1.1. LivDet 2015 Dataset

The LivDet 2015 [7] dataset comprises fingerprint images from four distinct scanners: GreenBit, Digital Persona, CrossMatch, and Biometrika. It contains more than 16,000 fingerprint images in total. In the training set, each scanner set contains approximately 1000 positive- and 1000 negative-labeled images. The testing set comprises 1000 positive- and 1500 negative-labeled images, of which an additional 500 are fabricated using materials that were not present in the training set. To validate the generalization capability of the model, the testing set contains new spoof materials (Liquid Ecoflex and RTV) that are not present in the training set.

#### 5.1.2. LivDet 2017 Dataset

The LivDet 2017 dataset [10], one of the most recent publicly available datasets, encompasses over 17,500 fingerprint images. These images were acquired using three distinct scanners: specifically, GreenBit, Orcanthus, and Digital Persona. The dataset also features separate training and testing sets for each scanner. The training set for each scanner consists of 1000 live and 1200 spoof fingerprint images, whereas the testing set encompasses over 1600 live and 2000 spoof images. A unique aspect of this dataset is the inclusion of additional spoof fingerprints fabricated using new materials such as gelatin, latex, and Liquid Ecoflex, which are not present in the training set. This inclusion aims to evaluate the generalization capabilities of the trained models: a crucial factor for practical applications of fingerprint spoof detection and real-world deployment of the trained model. Detailed attributes of the dataset are summarized in Table 3.

### 5.2. Ensemble Enhancement

For the ensemble, we used our Spoof Detection network along with *n* number of networks trained with our proposed methodology. The input to the ensemble network is a fingerprint image of size 1×512×512, which produces n+1 response maps that are stacked together to generate the final response map of size (n+1)×26×26. The mean value of the produced response map is then utilized to obtain the final prediction. Our ensemble network has demonstrated a substantial and consistent increase in accuracy across all the sensors, as depicted in Figure 9. We were able to achieve accuracies of 95.08%, 99.19%, 98.44%, and 99.04% for Digital Persona, CrossMatch, Biometrika, and GreenBit sensors, respectively, on the LivDet 2015 dataset. The average accuracy across all the sensors increased from the previous best of 96.19% to 97.94%, as shown in Table 4. The ensemble network achieved an average increase in accuracy of 0.4% compared to a single network, and our proposed method achieved an overall increase in accuracy of 1.75% compared to the previous state-of-the-art results.

We also were able to achieve accuracies of 94.92%, 94.56%, and 96.57% for the Digital Persona, Orcanthus, and GreenBit senors, respectively, on the LivDet 2017 dataset, as presented in Table 5. Our proposed ensemble method achieved an average classification accuracy of 95.35%, which is comparable to the 95.25% produced by the existing state-of-the-art method.

The achieved ensemble accuracies resulted from the selection of an optimal number of ensemble members for the network. To conduct this experiment, we gradually incorporated ensemble members into the network and observed the corresponding changes to prediction accuracy, as illustrated in Figure 9. Subsequently, we employed these members to create an ensemble network for each scanner. The solid blue lines correspond to the network accuracy achieved with the proposed method, while the orange and green dashed lines correspond to the network accuracy obtained with the traditional ensemble and snapshot ensemble methods, respectively [4]. As evident from this figure, the proposed method is able to increase or maintain accuracy, even when the accuracy is saturated with a base network. The conventional ensemble approach shows an immediate drop in the accuracy as soon as an additional network is employed. However, the illustration also indicates a notable trend: beyond the inclusion of four ensemble members, there appears to be some lose in accuracy. In light of this observation, we determined that the optimal number of ensemble members for our network is n=3+1, which equates to a total of 4 members. This decision is based on the empirical evidence suggesting that this configuration yields the most favorable balance between accuracy and the number of ensemble members. Consequently, all reported accuracies in our study are based on a network configuration utilizing four ensemble members.

The loss of accuracy with a higher number of ensemble members can be attributed to the inherent limitations of the fingerprint dataset, which is relatively small in size and, therefore, consists of very few good local minima. Our CAM-guided training procedure forces the models to converge in different local minima, and hence, some of the models have higher error rates than others. Ensembling more models with higher error rates results in degradation of network accuracy. Nevertheless, it is noteworthy that the impact of fewer good local minima is significantly less pronounced for our method compared to previous methodologies.

### 5.3. Comparison with Previous Methods

The network is trained to label the spoof features as −1 and the live features as +1. Therefore, computing the mean value of the response map indicates the network’s assumption of the input image as a live or spoof fingerprint. If the mean of the generated response map is negative, then the image is classified as ’spoof’, and if the mean value is positive, then the image is classified as ’live’. With the LivDet 2015 dataset, our proposed network architecture achieved accuracies of 94.24%, 99.19%, 97.92%, and 98.88% for the Digital Persona, CrossMatch, Biometrika, and GreenBit sensors, respectively. The proposed approach outperformed the previous state-of-the-art results for all scanners except the CrossMatch scanner, as demonstrated in Table 4. Although our proposed method does not achieve state-of-the-art results for the CrossMatch scanner, it does enhance the accuracy of the network.

### 5.4. Enhancement in Robustness

The LivDet 2015 testing set consists of 500 fabricated fingerprints for each sensor that were not included in the training set. To assess the cross-material robustness of the network, we evaluated our proposed method using these data, and the results are presented in Table 6. As there were no new materials present in the testing set for the CrossMatch sensor, it was excluded from our evaluation. Our tests demonstrated consistent and significant improvement in cross-material robustness. On average, we outperformed the current state-of-the-art results by 1.07%. Although we were unable to achieve state-of-the-art results for the Digital Persona sensor, our proposed method has been shown to steadily increase the accuracy of a single network across all the sensor data, resulting in significant enhancement of the robustness of the Spoof Detection network.

### 5.5. Feature-Space Analysis

To assess the efficacy of our training methodology in distinguishing features, we conducted a feature-space analysis using the LivDet 2015 testing set for all four scanners. This involved producing two-dimensional t-SNE feature embeddings of both live and spoof fingerprints. The features were extracted from the final convolutional layer of the Spoof Detection CNN; we used approximately 2500 samples from each scanner. The resultant t-SNE feature embeddings are depicted in Figure 10 and demonstrate notable separation between live and spoof features.

A sensor-specific analysis reveals nuanced differences. For instance, the feature embeddings of the Digital Persona sensor exhibit closer proximity between live and spoof features, resulting in a comparatively lower cross-material robustness accuracy with our method. Conversely, for scanners such as GreenBit, CrossMatch, and Biometrika, the feature embeddings are more distinctly separated, leading to a cross-material robustness accuracy that outperforms existing state-of-the-art methodologies, as presented in Table 6. The results derived from this analysis provide compelling evidence supporting the effectiveness of our proposed method in terms of generalization capabilities and cross-material robustness as a spoof detection network.

### 5.6. Ablation Study

To evaluate the influence of varying threshold values on the efficacy of our proposed training methodology, we conducted an experiment using the LivDet 2017 dataset. This experiment was designed to investigate how different threshold settings affect the performance of the proposed spoof detection network.

Initially, we trained the spoof detection network for each of the three sensors included in the LivDet 2017 dataset. This training was conducted in accordance with the protocols detailed in Section 4 and resulted in what we refer to as our ‘base models’. Subsequently, we applied our proposed training method to these base models, altering only the threshold values. The objective was to observe how these modifications impact the model’s accuracy. The outcomes of these experiments are presented in Table 7.

A key observation is that the use of threshold values of t=0.9 and t=0.8 yields higher accuracy compared to the base models. However, setting the threshold value below 0.8 leads to a reduction in network accuracy. This decline is attributed to the loss of essential features in the fingerprint images, as demonstrated in Figure 7. Based on these results, it becomes evident that the optimal threshold value for training with our proposed method is t=0.9 using the current settings. This threshold level provides balance by effectively obscuring the most-active regions of the fingerprint image while preserving all other essential features necessary for accurate classification. We were able to achieve classification accuracies of 94.70%, 96.17%, and 94.19% for the Digital Persona, GreenBit, and Orcanthus sensors, respectively, using this threshold value.

### 5.7. Computational Requirements

The proposed Spoof Detection CNN is a relatively small model in terms of parameters and computation time. The model comprises 29,633 parameters in total and requires $2.83\times 10^{8}$ FLOPs for inference. For training, it takes around 3 h to train a single model using our methodology on an NVIDIA 4090 GPU. Furthermore, the average classification time for a single input image on the same GPU is 0.28 ms, and on an Intel Core i9-10940X @ 3.30GHz CPU, it is 23.86 ms. The model can differentiate between live and spoof fingerprints at a very high speed. The blend of minimal parameter size and rapid classification speed of the model underscores its suitability and practicality for deployment in various real-world scenarios requiring spoof fingerprint detection.

## 6. Conclusions

We introduce a simple training procedure that uses activation perturbation of a network to obtain enhanced ensembles of neural networks and improved performance of spoof fingerprint detection with a small training dataset by incorporating a class activation map in the training procedure. We demonstrate that our training procedure can produce diverse ensemble members by simply adding noise to the CAM regions. We show in our experiments that our approach improves the state-of-the-art results for spoof fingerprint detection. Moreover, we demonstrate improved robustness for a spoof fingerprint detection model with our proposed training procedure.

The proposed method, while effective, may have the following limitations:The optimal number of ensemble members may vary depending on the specific characteristics of the network architecture in use. What is effective for one architecture might not yield the same level of accuracy or efficiency for another.The optimal threshold value for the activation map might differ when the method is applied to network architectures different from the one used in our study. Factors such as the diversity of spoofing materials and sensor types can also influence the ideal threshold setting.

Future research endeavors will be directed towards investigating the application of activation perturbation and ensemble learning techniques. The primary objective of this exploration is to enhance the cross-sensor efficacy of spoof fingerprint detection networks. This enhancement aims to facilitate the development of models with superior generalization capabilities. Activation perturbation will be employed to bolster the robustness of the network against sensor-specific variations, while the integration of ensemble learning strategies is anticipated to aggregate the strengths of diverse models. Consequently, this synergistic approach is expected to yield significant improvements in network performance across various sensor types, thereby addressing one of the critical challenges in the domain of biometric security. 

## Figures and Tables

**Figure 1 sensors-24-00187-f001:**
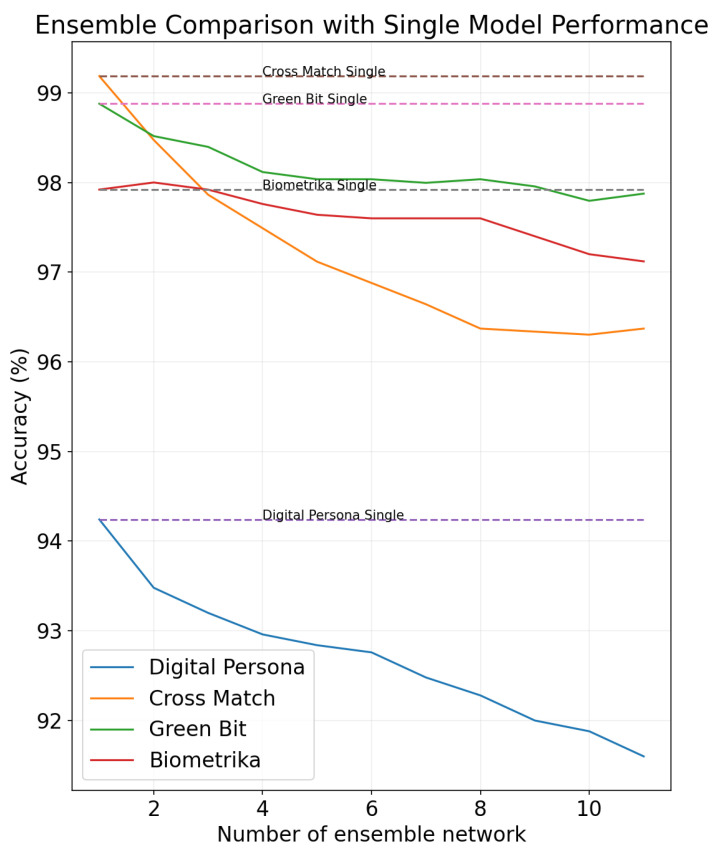
Performance variation of a CNN-based spoof detection network when subjected to an ensemble setting.

**Figure 2 sensors-24-00187-f002:**
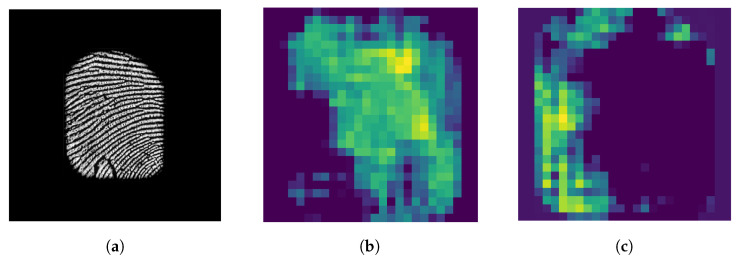
Impact of the proposed training method on the activation patterns of a CNN: (**a**) input fingerprint, (**b**) activation of the base network, and (**c**) activation of one of the ensemble members after being trained with the proposed method.

**Figure 3 sensors-24-00187-f003:**
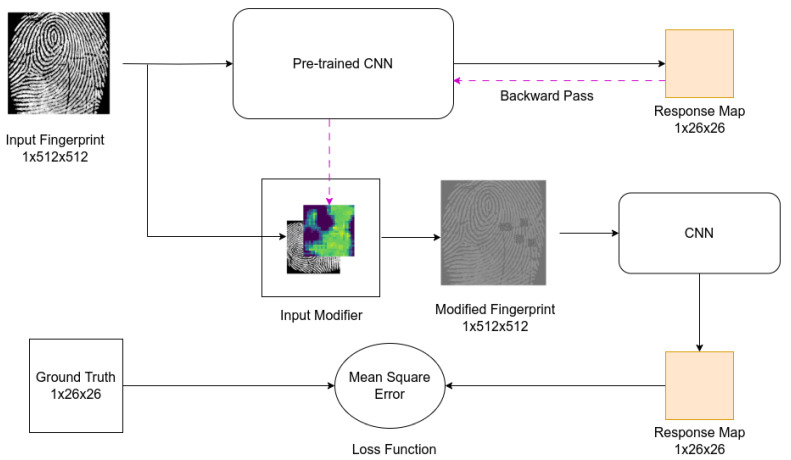
Visualization of our proposed training procedure.

**Figure 4 sensors-24-00187-f004:**
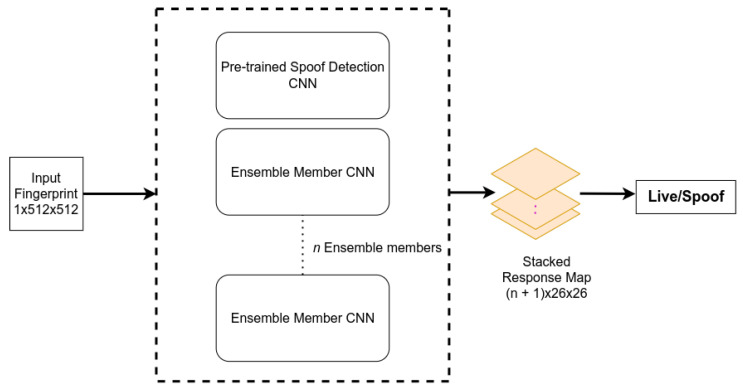
Proposed prediction method of the ensemble network. The final prediction is achieved by stacking the response maps of all the ensemble members.

**Figure 5 sensors-24-00187-f005:**
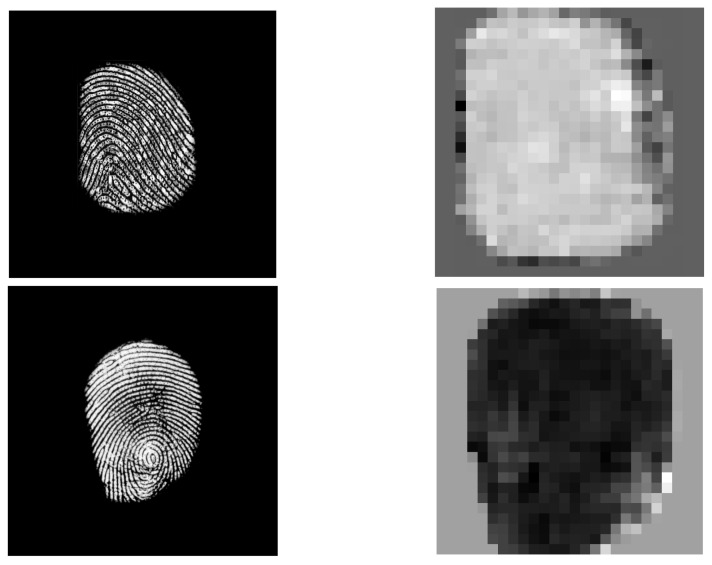
Input and response map of the proposed CNN. The top row shows the live fingerprint image and its response, while the bottom row presents the spoof fingerprint image and its response.

**Figure 6 sensors-24-00187-f006:**
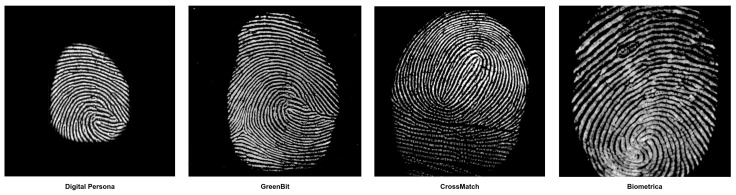
Fingerprint images from all the sensors from LivDet 2015 dataset after pre-processing steps.

**Figure 7 sensors-24-00187-f007:**
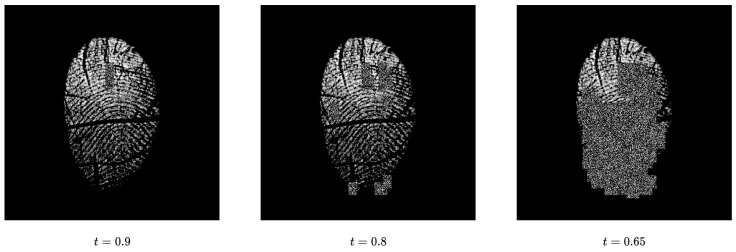
Impact of different threshold values on input image for imposing noise using activation map. The presented image corresponds to the training set of CrossMatch sensor from LivDet 2015 dataset.

**Figure 8 sensors-24-00187-f008:**
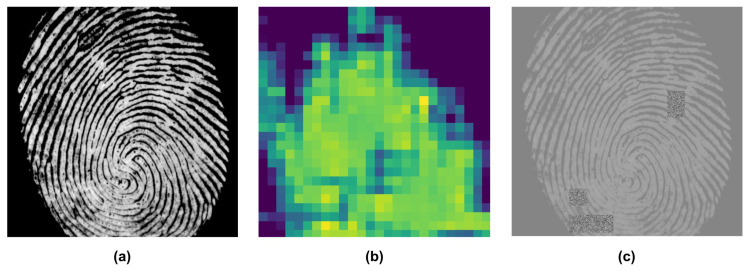
Noise addition process using activation map: (**a**) original input image, (**b**) activation map of the input image, and (**c**) modified input image.

**Figure 9 sensors-24-00187-f009:**
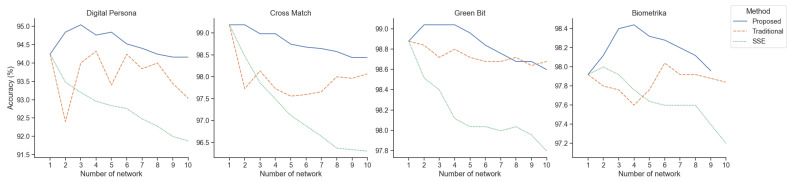
Change in accuracy with respect to the number of ensemble members for all four scanners in LivDet 2015 dataset.

**Figure 10 sensors-24-00187-f010:**
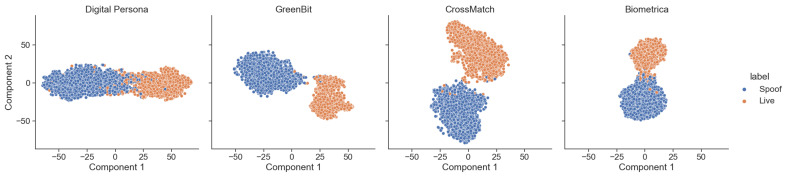
Two-dimensional t-SNE feature embeddings of the Spoof Detection network trained with the proposed method and tested on LivDet 2015 testing set for all four sensors.

**Table 1 sensors-24-00187-t001:** Spoof Detection CNN architecture details.

Layer	Output Size	Stride	Kernel
Input image	1×512×512	–	–
Convolution	32×253×253	2×2	7×7
MaxPool	32×126×126	2×2	2×2
ReLU	32×126×126	–	–
Convolution	32×124×124	1×1	3×3
MaxPool	32×62×62	2×2	2×2
ReLU	32×62×62	–	–
Convolution	32×60×60	1×1	3×3
MaxPool	32×30×30	2×2	2×2
ReLU	32×30×30	–	–
Convolution	32×28×28	1×1	3×3
ReLU	32×28×28	–	–
Convolution	1×26×26	1×1	3×3
**Total Parameters**			29,633

**Table 2 sensors-24-00187-t002:** Summary of studies focused on fingerprint spoof detection using CNNs.

Method	Approach	Database	Performance
Emanuela et al. [40]	Employment of Siamese network	LivDet 2013	Avg. Accuracy = 93.1%
Menotti et al. [39]	Combination of hyper-parameter tuning of a CNN and use of SVM for prediction	LivDet 2015	Avg. Accuracy = 93.745%
Nogueira et al. [35]	Transfer learning using VGG network	LivDet 2015	Avg.Accuracy = 95.5%
Jung et al. [38]	Liveness detection of probe fingerprint using template fingerprint through two sequential custom CNNs	LivDet 2015	Avg. Accuracy = 96.99%
Chugh et al. [42]	Minutiae-centered local patch extraction and detection through MobileNet	LivDet 2011-2015	ACE = 1.48% (LivDet 2015)
Zhang et al. [41]	Slim-ResCNN and patch extraction via center of gravity	LivDet 2017	Avg. Accuracy = 95.25%
Chugh et al. [44]	Style transfer between known spoof materials to generate synthetic data for network training	LivDet 2017	Avg. Accuracy = 95.88%
Liu et al. [31]	Fusion of global and local spoofness score and patch extraction using Grad-CAM during inference	LivDet 2017	TDR = 91.19% @FDR = 1%
**Proposed Approach**	Ensemble of CAM-guided models generated from a pre-trained CNN	LivDet 2015	Avg. Accuracy = 97.94%

**Table 3 sensors-24-00187-t003:** Summary of the datasets used in this study.

Dataset	LivDet 2015 [7]				LivDet 2017 [10]		
**Scanner**	Digital Persona	GreenBit	CrossMatch	Biometrika	Digital Persona	GreenBit	Orcanthus
**Image Size**	252×324	500×500	640×480	1000×1000	252×324	500×500	$300\times n{\;}^{*}$
**Resolution** (dpi)	500	500	500	1000	500	569	500
**Live Images (Train/Test)**	1000/1000	1000/1000	1510/1500	1000/1000	999/1692	1000/1700	1000/1700
**Spoof Images (Train/Test)**	1000/1500	1000/1500	1473/1448	1000/1500	1199/2029	1200/2040	1200/2018
**Known Spoof Materials (Training)**	Ecoflex, Gelatin, Latex, Wood Glue				Wood Glue, Ecoflex, Body Double		
**Unknown Spoof Materials (Testing)**	Liquid Ecoflex, RTV				Gelatin, Latex, Liquid Ecoflex		

* Fingerprint images captured using Orcanthus scanner have variable heights (350–450 px).

**Table 4 sensors-24-00187-t004:** Accuracy of the proposed method compared with previous state-of-the-art CNN methods along with some LivDet 2015 competition methods. Our proposed method exhibits consistent increases in accuracy across all the sensors.

Scanner	Digital Persona	CrossMatch	Biometrika	GreenBit	Avg. Accuracy
Image Size	252×324	800×750	1000×1000	500×500	
hbirkholz [7]	88.0	89.93	93.40	91.36	90.60
titanz [7]	89.04	91.62	92.36	91.76	91.20
jinglian [7]	88.16	94.34	94.08	94.44	92.76
unina [7]	85.44	96.00	95.20	95.80	93.11
CNN-VGG [35]	93.72	98.10	94.36	95.40	95.40
Gram-128-CNN [49]	91.5	99.73	95.90	98.65	96.45
LM-CNN [38]	91.92	99.15	96.72	96.96	96.19
Proposed Spoof Detection CNN	94.24	99.19	97.92	98.88	97.56
Proposed Ensemble	95.08	99.19	98.44	99.04	97.94

**Table 5 sensors-24-00187-t005:** Comparison of classification accuracy of the proposed method with one of the SOTA methods utilizing LivDet 2017 dataset.

Sensor	Slim-ResCNN [41]	Spoof Detection CNN	Proposed Ensemble
Digital Persona	95.59	94.7	94.92
Orcanthus	93.71	94.19	94.56
GreenBit	96.44	96.17	96.57
**Average**	95.25	95.02	95.35

**Table 6 sensors-24-00187-t006:** Cross-material robustness comparison of the proposed method on LivDet 2015 dataset [7]. The proposed method was tested on materials used for generating spoof fingerprints that were not present in the training dataset.

Training Materials	Testing Materials	Sensor	SDCNN	SlimRCNN [41]	FSB [42]	Proposed Ensemble
Ecoflex, Gelatin, Latex, Wood Glue	Liquid Ecoflex, RTV	Digital Persona	95.4%	93.60%	98.2%	97.8%
Ecoflex, Gelatin, Latex, Wood Glue	Liquid Ecoflex, RTV	GreenBit	98.8%	96.89%	98.2%	99.4%
Ecoflex, Gelatin, Latex, Wood Glue	Liquid Ecoflex, RTV	Biometrika	99.6%	92.28%	97.4%	99.8%
**Average**			97.93%	94.26%	97.93%	99.0%

SDCNN = Spoof Detection CNN (proposed); SlimRCNN = SlimResNet CNN; FSB = Fingerprint Spoof Buster.

**Table 7 sensors-24-00187-t007:** Ablation study using different values of threshold (t) on LivDet 2017 dataset.

Sensor	Base Model	t=0.9	t=0.8	t=0.65
Digital Persona	93.95	94.70	93.68	92.58
GreenBit	95.5	96.17	95.88	54.70
Orcanthus	94.03	94.19	94.08	87.84

## Data Availability

Publicly available datasets are utilized in this study. The datasets can be obtained from: https://livdet.org/registration.php (accessed on 16 June 2023).

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
