# Peer review of "Enhancing Ensemble Learning Using Explainable CNN for Spoof Fingerprints"

_sensors, 2023, doi:10.3390/s24010187_

Round 1

Reviewer 1 Report

Comments and Suggestions for Authors

The manuscript focuses on Enhancing Ensemble Learning using Explainable CNN for Spoof Fingerprints. While the research work is interesting, it still faces several technical limitations that obstruct a clear understanding of the primary theme. Some of the major limitations are outlined below:

1.      How well does the proposed method address the interpretability issues associated with Convolutional Neural Networks (CNNs) in the context of fingerprint spoof detection?

2.      Can the authors elaborate on the significance of using Class Activation Map (CAM) in concealing specific fingerprint regions during training, and how it contributes to achieving the same objective from different regions in subsequent networks?

3.      What evidence is provided to support the claim that the proposed method enhances classification accuracy and robustness across multiple sensors, and how was the evaluation conducted on the LivDet dataset?

4.      In the introduction, the authors mention the trade-offs associated with ensemble learning. Could the authors provide more details on these trade-offs and how the proposed method mitigates them?

5.      The paper refers to the LivDet dataset for evaluation. What are the characteristics of this dataset, and how representative is it of real-world scenarios for spoof fingerprint detection?

6.      How does the proposed training method compare to existing ensemble techniques, especially those mentioned in the introduction, in terms of training cost, accuracy improvement, and robustness?

7.      The paper discusses the use of cyclic cosine annealing learning rate and Stochastic Weight Averaging. Can the authors elaborate on how these techniques contribute to the diversity of ensemble members and why they chose them for their approach?

8.      The authors mention biases in small and homogeneous datasets. How does the proposed method address biases in the training dataset, and what impact do biases have on ensemble accuracy?

9.      How does the integration of the Gradient-weighted Class Activation Map (Grad-CAM) help in visualizing internal representations learned by the CNN, and how does it contribute to the proposed ensemble method?

10.  In Figure 2, the paper illustrates activation perturbation. Can the authors provide more details on how this perturbation is generated and its impact on generating diverse models within the ensemble?

11.  The paper mentions the small size of datasets for training spoof detection networks. How does the limited size of the dataset impact the generalizability of the proposed method to real-world scenarios?

12.  Could the authors elaborate on the choice of network architecture (Spoof Detection CNN) presented in Table 1 and how it contributes to the effectiveness of the proposed method?

13.  How does the proposed method ensure different converging points by using the activation map to hide selected regions of the input, and what is the impact on the diversity of the resulting ensemble networks?

14.  Please provide the parameters and time complexity of the proposed model. Also, show the tSNE plot to validate the discriminative nature of the extracted features.

15.  Please add the comparative analysis table of discussed studies in the related work section.

16.  Please discuss any limitations or potential challenges associated with the proposed method, especially in real-world applications of spoof fingerprint detection.

Comments on the Quality of English Language

minor.

Author Response

Dear Reviewer and Editor.

The detailed responses are attached as pdf file. We hope we were able to make appropriate responses to the questions and comments.

Naim Reza and Ho Yub Jung

Reviewer 2 Report

Comments and Suggestions for Authors

Very interesting contribution, but to improve correctness and completeness, there are some concerns to be addressed.

Comment 1

In the literature, it has been shown that different CAM methods produce conflicting results[https://doi.org/10.1101/2021.12.23.21268289, https://doi.org/10.1109/ACCESS.2023.3327808]. Indeed, these methods enable just a local explanation for a specific instance (i.e., fingerprint), not allowing a global validation of the system prediction. For these reasons, saliency maps have still to demonstrate to be an objective tool for validating clinical findings. Considering that the explanation is the focus of this work, it is fundamental to discuss this aspect in the 'Introduction' section.

Comment 2

Limitations of the study have to be highlighted in the 'Conclusion' section.

Comment 3

In Figure 3, for completeness, it is necessary to show at least an image sample for each fingerprint scanner (sensor).

Comment 4

It should be interesting to make an evaluation concerning the optimal choice for the 'number of ensemble', regardless of the specific fingerprint scanner.

Comments on the Quality of English Language

An overall English revision could be appropriate, to improve the fluency of the language.

Author Response

(The authors gave the same response as above.)

Reviewer 3 Report

Comments and Suggestions for Authors

The presentation of method is not clear enough, as detailed below:

1. Since the sizes of output and image is different, how to map the output information to the corresponding position in the input image.

2. Please provide the description of the ground-true.

3. Please provide the details of the feature map activation of each class.

4. How to set the threshold on the generated map.

5.In the experiments, how many CNNs are ensembled for testing.

6. Maybe a quantitative evaluation on the diversity of ensemble learning is needed.

7. Only one dataset is tested, it is hard to support robustness. And some important parameters lack comprehensive discussion.

Author Response

(The authors gave the same response as above.)

Reviewer 4 Report

Comments and Suggestions for Authors

1.The title of the paper is "Enhancing Ensemble Learning using Explainable CNN for Spoof Fingerprints", but there does not seem to be an introduction to the interpretability of CNN methods. Please add clarification on this.

2.In Section 3 of the article, "Methodology", use the threshold “t” to identify the more active regions in the activation graph. The experimental part “t” is set to 0.9, but the determination of the threshold lacks relevant experimental analysis in the paper, please make relevant additions.

3.In the process of network training, how to obtain Ground Truth for training the size of 1*26*26 is not explained in detail in the article, please add this.

4.Please explain that the accuracy of the comparison method used in Table 2 of the comparison experiment results in the article does not seem to be the same as the accuracy in the original text.

5.In Table 3 of the experimental part, the cross-material robustness comparison compares only one method except for the comparison itself, which may not represent the most advanced research results, please add the comparison method appropriately.

Comments on the Quality of English Language

Minor editing of English language required

Author Response

(The authors gave the same response as above.)

Round 2

Reviewer 1 Report

Comments and Suggestions for Authors

The authors have adeptly incorporated the previous feedback, resulting in substantial enhancements to the paper. 

Reviewer 2 Report

Comments and Suggestions for Authors

All comments have been properly approached.

Reviewer 3 Report

Comments and Suggestions for Authors

No other suggestions.